# Dopamine D_2_ Receptor Agonist Binding Kinetics—Role of a Conserved Serine Residue

**DOI:** 10.3390/ijms22084078

**Published:** 2021-04-15

**Authors:** Richard Ågren, Tomasz Maciej Stepniewski, Hugo Zeberg, Jana Selent, Kristoffer Sahlholm

**Affiliations:** 1Department of Neuroscience, Karolinska Institutet, 171 77 Stockholm, Sweden; hugo.zeberg@ki.se; 2Research Programme on Biomedical Informatics (GRIB), Hospital del Mar Medical Research Institute (IMIM)—Pompeu Fabra University (UPF), 08003 Barcelona, Spain; tm.stepniewski@gmail.com (T.M.S.); jana.selent@upf.edu (J.S.); 3InterAx Biotech AG, 5234 Villigen, Switzerland; 4Faculty of Chemistry, Biological and Chemical Research Centre, University of Warsaw, 02-089 Warsaw, Poland; 5Department of Integrative Medical Biology, Wallenberg Centre for Molecular Medicine, Umeå University, 901 87 Umeå, Sweden

**Keywords:** *Xenopus* oocytes, electrophysiology, voltage-clamp, molecular dynamics simulation, G protein-coupled receptor, phenethylamines, aminotetralins

## Abstract

The forward (k_on_) and reverse (k_off_) rate constants of drug–target interactions have important implications for therapeutic efficacy. Hence, time-resolved assays capable of measuring these binding rate constants may be informative to drug discovery efforts. Here, we used an ion channel activation assay to estimate the k_on_s and k_off_s of four dopamine D_2_ receptor (D_2_R) agonists; dopamine (DA), p-tyramine, (R)- and (S)-5-OH-dipropylaminotetralin (DPAT). We further probed the role of the conserved serine S193^5.42^ by mutagenesis, taking advantage of the preferential interaction of (S)-, but not (R)-5-OH-DPAT with this residue. Results suggested similar k_off_s for the two 5-OH-DPAT enantiomers at wild-type (WT) D_2_R, both being slower than the k_off_s of DA and p-tyramine. Conversely, the k_on_ of (S)-5-OH-DPAT was estimated to be higher than that of (R)-5-OH-DPAT, in agreement with the higher potency of the (S)-enantiomer. Furthermore, S193^5.42^A mutation lowered the k_on_ of (S)-5-OH-DPAT and reduced the potency difference between the two 5-OH-DPAT enantiomers. Kinetic K_d_s derived from the k_off_ and k_on_ estimates correlated well with EC_50_ values for all four compounds across four orders of magnitude, strengthening the notion that our assay captured meaningful information about binding kinetics. The approach presented here may thus prove valuable for characterizing D_2_R agonist candidate drugs.

## 1. Introduction

G protein-coupled receptors (GPCRs) are ubiquitously expressed in the human body and targeted by over 30% of all FDA-approved drugs [1]. Recently published structures of catecholamine receptors, which belong to the rhodopsin-like GPCRs (also known as class A [1]), are consistent with important polar interactions between orthosteric agonist amine groups and a conserved aspartate (D^3.32^) in the transmembrane segment (TM) 3 and between agonist electronegative groups (such as hydroxyls) and three conserved serine (S) residues in TM) 5; S^5.42^, S^5.43^, and S^5.46^ [2,3,4] (superscript numbers represent the Ballesteros–Weinstein numbering scheme [5]). Dopamine receptors form a subgroup of catecholamine receptors and are further divided into D_1_-like (D_1_ and D_5_) and D_2_-like (D_2_, D_3_, and D_4_) receptors based on sequence homology and G protein coupling specificity; the former being coupled mainly to stimulatory G_s_ proteins whereas the latter signal through inhibitory G_i/o_ proteins [6]. In addition to classical G protein-dependent pathways evoking adenylate cyclase inhibition, potassium channel activation, and calcium channel inhibition, activated D_2_-like receptors are known to elicit a number of other downstream responses, such as extracellular signal-regulated kinase (ERK) phosphorylation and Akt dephosphorylation, the latter of which is induced via beta-arrestin2 recruitment to the receptor [7]. The dopamine D_2_ receptor (D_2_R) functions as an autoreceptor on dopaminergic terminals and is found postsynaptically in many brain regions, notably including the striatum, where it is prominently expressed on medium spiny neurons and forms heteromers with several other GPCRs, such as adenosine A_2A_ receptors [8]. D_2_R has been implicated in several physiological processes such as locomotion, working memory and reward learning, and the regulation of prolactin secretion. In agreement, D_2_R is an important target for treating neurological, psychiatric, and endocrine disorders, including Parkinson’s disease, schizophrenia, and hyperprolactinemia [6,7]. Dopamine (DA) binding affinity and functional potency at D_2_R has been demonstrated to be highly dependent on S193^5.42^, as alanine (A) mutation of this residue reduces DA affinity and potency by 80- to 800-fold [9,10,11,12,13]. In contrast, the affinity of the weak partial agonist, p-tyramine, which lacks the meta-hydroxyl of dopamine (Figure 1A), is decreased less than twofold by the same mutation [9,14]. Certain synthetic, structurally constrained agonists, such as monohydroxylated *N*,*N*-dipropyl-2-amino-tetralins (DPATs), are believed to interact with only one of the TM5 serine residues in a stereospecific manner. For example, (S)-5-OH-DPAT has been postulated to form an important hydrogen bond with S193^5.42^, whereas its stereoisomer, (R)-5-OH-DPAT, instead prefers interacting with S197^5.46^ [14,15]. Interestingly, S193^5.42^A mutation has been found to have a differential impact on the efficacies of certain synthetic agonists depending on the signaling pathway examined [11].

The network of contacts between the binding pocket and the ligand governs ligand affinity by determining the association (k_on_) and dissociation rate constants (k_off_) of the ligand–receptor complex. Interestingly, agonist binding kinetics may influence the efficacy and downstream signaling preferences (bias) [16,17,18]. In addition, the kinetics of ligand binding has garnered increasing attention as a way to optimize in vivo receptor occupancy, ensuring therapeutic efficacy and avoiding drug-induced side effects [19,20]. Agonist binding kinetics are thus of interest both for therapeutic ligand design and from a basic science perspective. Many GPCRs, including D_2_R, are known to exist in two distinct affinity states with regard to agonist binding. In general, the high-affinity state is favored by the binding of a G protein to the receptor and is considered to be the functional, signaling state of the receptor, whereas the low-affinity state predominates in the absence of bound G protein [21,22]. Previous investigations of aminergic GPCRs have investigated the kinetics of agonist binding to isolated membranes [17,23], measured agonist-induced incorporation of radiolabeled guanine nucleotide analogues [24], and the activation rates of receptors and G proteins using fluorescently labeled constructs [25,26].

To our knowledge, estimates of association (k_on_) and dissociation (k_off_) rate constants for D_2_R agonists at the G protein-coupled signaling state have not been previously reported from live cells. Thus, in the present investigation, we examined (S)- and (R)-5-OH-DPAT, DA, and p-tyramine using G protein-coupled inward rectifier potassium (GIRK) channel currents in *Xenopus* oocytes as readout. GIRK channels are opened by Gβγ subunits released from G_i/o_ protein trimers activated by agonist-stimulated GPCRs [27]. To examine the role of S193^5.42^ for k_on_ and k_off_ of these D_2_R agonists, both WT and S193^5.42^A mutant D_2_R constructs were used. k_on_ and k_off_ were estimated from the time courses of activation and deactivation of agonist-induced GIRK currents upon agonist application and washout, respectively. The experimental data was also compared with molecular dynamics simulations based on the active-state structure of D_2_R in complex with an inhibitory G protein and bound to the agonist, bromocriptine [3]. Our results suggest that contacts with S193^5.42^ have a relatively greater impact on k_on_, compared to k_off_, for both DA and (S)-5-OH-DPAT, whereas the binding kinetics of p-tyramine and (R)-5-OH-DPAT were only marginally affected by S193^5.42^A mutation. In agreement, the two enantiomers of 5-OH-DPAT were distinguished mainly by their k_on_ rates at the WT D_2_R and displayed similar kinetics at the S193^5.42^A mutant receptor.

## 2. Results

### 2.1. Phenethylamine and DPAT Potencies and Efficacies at WT and S193^5.42^A D_2_R

Following agonist application to Xenopus oocytes expressing D_2_R and GIRK1/4 subunits, G_βγ_ subunits released from activated G_i/o_ protein heterotrimers open the GIRK channels. Co-expression of the regulator of G protein signaling-4 (RGS4) accelerates the GTP hydrolysis rate at the Gαi/o subunit and increases the G protein cycle’s turnover rate, such that the time course of the GIRK current more closely follows the time course of D_2_R agonist occupancy [27,28]. Here, we used the GIRK current in oocytes voltage-clamped at −80 mV and superfused with a high-K^+^ buffer (25 mM KCl) as a readout of agonist binding to D_2_R.

Thus, increasing agonist concentrations were applied consecutively to voltage-clamped oocytes expressing WT D_2_R, RGS4, and GIRK1/4. In each oocyte, the elicited current responses were normalized to the response to a maximally effective concentration of DA (1 µM) to yield concentration-response relationships. To avoid oocyte deterioration during data acquisition, as well as to minimize the potential effects of current rundown [29] or receptor desensitization on the results, concentration–response data was acquired for only one agonist per oocyte. A representative trace of currents measured during p-tyramine concentration–response data acquisition is shown in Appendix A). The tested agonists displayed the following rank order of potencies: (S)-5-OH-DPAT > DA > (R)-5-OH-DPAT > p-tyramine (Figure 1A,B), in agreement with previously published data on these agonists [14,30]. In oocytes expressing S193^5.42^A mutant D_2_R, RGS4, and GIRK1/4, responses were normalized to the response to 300 µM DA (concentrations of 1 mM and above were found to block GIRK channels; data not shown) and the rank order of agonist potencies were: (R)-5-OH-DPAT > (S)-5-OH-DPAT > DA > p-tyramine (Figure 1C). The differences in rank order of potency between WT and S193^5.42^A mutant D_2_R were driven mainly by the 467- and 52-fold reductions in DA and (S)-5-OH-DPAT potency at S193^5.42^A (Figure 1D). The DA-normalized agonist efficacies increased 5.7- and 4.1-fold for (R)-5-OH-DPAT and p-tyramine, respectively, at S193^5.42^A (Figure 1E). The strategy of acquiring concentration-response data for only one agonist per oocyte is unlikely to have had much impact on the estimation of relative efficacies since receptor reserve is low or absent under the conditions used [28,31]. In agreement with this notion, when averaging data across all experiments (Appendix A), the relative current response amplitudes evoked by maximally effective concentrations of DA, and the three partial agonists via WT and mutant receptors, are similar to the relative efficacies shown in Figure 1E.

### 2.2. (R)- and (S)-5-OH-DPAT Dissociation Rates Are Similar, But Slower Than Those of DA and p-Tyramine at WT D_2_R

The deactivation time course of agonist-induced GIRK currents upon agonist washout has been shown to reflect agonist residence time at the receptor; i.e., 1/k_off_ [32,33]. We obtained such estimates of k_off_ by analyzing the washout phase following short (13 s) applications of submaximally effective agonist concentrations (see legend of Figure 2) in oocytes co-expressing WT D_2_R with GIRK1/4 and RGS4. GIRK responses elicited by both (R)- and (S)-5-OH-DPAT decayed with a similar time course upon washout, but slower than responses elicited by DA and p-tyramine (Figure 2A,D). To test the notion that the slower decay kinetics of GIRK responses to (R)- and (S)-5-OH-DPAT were due to a longer residence time at the receptor rather than to accumulation of these lipophilic agonists in the oocyte membrane, we also measured GIRK response decay kinetics when agonist availability to the D_2_R was terminated by the addition of an antagonist in the continued presence of agonist. As can be seen in Figure 2B, the time courses of response decay upon addition of 1 µM of the D_2_R antagonist, haloperidol, were similar to the washout-induced decay rates for (R)- and (S)-5-OH-DPAT. As is also apparent, the DA-induced GIRK response decayed faster under these conditions as well (see Appendix A for quantification of the haloperidol-induced response decay kinetics). While haloperidol has been reported to block GIRK channels directly, thus potentially confounding the response decay rates measured here, this effect occurs with an IC*_50_* of 41 µM and channel block at 1 µM is negligible [34,35]. In agreement, we have verified that haloperidol at 3 µM induces only 5.9 ± 0.2% (*n* = 4) direct GIRK current block in our hands. The effect at the D*_2_*R receptor is much more potent and 1 µM haloperidol fully blocks the GIRK response to even 100 nM DA [28]. Thus, the effect of haloperidol reported here most likely reflects action at the D*_2_*R, rather than at the GIRK channel.

The time course of solution exchange around the oocyte under the conditions used here was also tested by switching from a low (1 mM) to a high (25 mM) K^+^ extracellular solution and monitoring the rate of change of basal GIRK current amplitudes. These experiments revealed a mean time constant of >2 s*^−^*^1^ (Appendix A); greater than the fastest rates of GIRK current deactivation and activation (see below) analyzed here. While the k_off_ values were in the same range for DA and p-tyramine at both WT D_2_R and the S193^5.42^A mutant, the mutation increased the decay rates of both (R)- and (S)-5-OH-DPAT-induced responses, such that they approached the k_off_s of DA- and p-tyramine (Figure 2C,D).

### 2.3. Estimated Association Rates of (R)- and (S)-5-OH-DPAT Differ at WT D_2_R but Are Similar at the S193^5.42^A Mutant

The nearly identical response deactivation time courses obtained with (R)- and (S)-5-OH-DPAT suggest that the higher affinity of (S)-5-OH-DPAT may result from a difference in k_on_, rather than in k_off_. We, therefore, investigated the time courses of GIRK current activation at different concentrations of agonist in oocytes co-expressing WT D_2_R or S193^5.42^A mutant D_2_R with RGS4 and GIRK1/4. At low agonist concentrations, evoking response time courses well below the solution exchange rate, the observed activation rate, k_obs_, was linearly dependent on the agonist concentration. At higher agonist concentrations, k_obs_ appeared to saturate (Appendix A), as previously observed for the rates of G protein- and receptor activation [25,36].

A straight line was fitted to the data over the linear range of the relation between k_obs_ and agonist concentration and the slope was taken as a measure of k_on_. As could be expected based on the previously obtained k_off_s and EC_50_s, the estimated k_on_s of (S)-5-OH-DPAT and DA were higher than those of (R)-5-OH-DPAT and p-tyramine. Compared to the WT receptor, k_on_ estimates for DA and (S)-5-OH-DPAT were reduced at the S193^5.42^A mutant (Figure 3A,B,D). In contrast, (R)-5-OH-DPAT, and to a lesser extent, p-tyramine, demonstrated slightly increased k_on_ values at the S193^5.42^A D_2_R (Figure 3B–D). Consequently, the k_on_ estimates of (R)- and (S)-5-OH-DPAT were similar at the S193^5.42^A mutant. To relate mutation-induced changes in estimated binding rates to the corresponding EC_50_ shifts (see Figure 1), kinetic K_d_ values were calculated from the ratios of k_off_ and k_on_ (Table 1). pEC_50_s for each agonist were plotted against the respective kinetic pK_d_ values, demonstrating a significant correlation (Figure 3E).

### 2.4. Structural Determinants of Mutation-Induced Changes in Compound Kinetics

In order to gain structural insight into how S193^5.42^A mutation impacts agonist–receptor interactions, we simulated the active conformation of D_2_R in complex with (R)-5-OH-DPAT, (S)-5-OH-DPAT, and DA. The binding mode of each compound was approximated by clustering the generated simulation frames based on the ligand coordinates and studying the most populated cluster. Both DA and (S)-5-OH-DPAT were observed to form simultaneous contacts with S193^5.42^ and histidine (H) 364^6.55^ (Figure 4A,B). Seeing as these two compounds formed extensive interactions with S193^5.42^, it is likely that altering the polar character of this residue would impair the binding of both compounds to D_2_R. In contrast to DA and (S)-5-OH-DPAT, (R)-5-OH-DPAT primarily forms polar interactions with S197^5.46^ and does not interact extensively with S193^5.42^ (Figure 4C). Indeed, simulating the studied compound within the S193^5.42^A D_2_R reveals that the mutation does not impact the primary binding mode of the compound. (Figure 4D).

## 3. Discussion

In the present study, we used the kinetics of agonist-induced response activation and deactivation in a GIRK channel-based electrophysiology assay to derive estimates of binding rate constants of four D_2_R agonists. We also aimed to investigate how agonist interactions with a conserved serine, S193^5.42^, in the D_2_R orthosteric binding pocket affect these rate constants. Since agonist k_on_ has often been assumed to be diffusion-limited [37,38], a particularly interesting finding is that the differential potencies/affinities of the two 5-OH-DPAT enantiomers appear to stem from differences in k_on_, rather than k_off_. The electrophysiology assay employed here has the advantage of using unmodified proteins and has been reported to demonstrate a very close temporal relation with G protein activation, as measured by fluorescence resonance energy transfer between labeled G protein subunits [39]. While this readout of agonist binding is an indirect one, our previous analysis of antipsychotic binding rates, also using GIRK currents as a proxy for D_2_R agonist occupancy, yielded results in very good agreement with radioligand- and fluorescence-based binding studies [28,40]. Similarly, other investigators have used oocyte-based GIRK assays to make inferences about the structural details of DA-D_2_R interactions [41,42].

However, we are aware that the temporal resolution of our assay is limited both by the rate of buffer exchange around the relatively large oocyte, as well as by the kinetics of G protein turnover. By co-expressing RGS4 and using high perfusion rates, the conditions of the washout-induced deactivation experiments were optimized to reflect agonist dissociation rates [12,27]. We also used conditions where there is little or no receptor reserve [28,43] in order for the concentration-response relationships of GIRK activation to more directly reflect D_2_R occupancy by agonist. In agreement with this notion, our measured EC_50_ value for DA (20 nM) is in good agreement with high-affinity site K_i_ values for DA binding to D_2_R reported from radioligand competition studies [14,44,45,46,47].

Here, we observed a linear dependence of GIRK current activation rates on agonist concentration in the lower, submaximally effective range, but not at higher concentrations (see Appendix A). Likewise, rates of α_2A_ receptor activation and G protein activation by muscarinic receptors were found to increase linearly at low, subsaturating agonist concentrations but showed a hyperbolic, saturating relationship at higher concentrations [25,36]. The maximal rates of G protein activation have also been found to differ between different agonists at the α_2A_ adrenergic receptor [48], presumably explaining differences in efficacy. Here, we deliberately used the linear range of the relation between k_obs_ and agonist concentration for k_on_ estimation.

Comparing agonist EC_50_s with kinetic K_d_ values calculated from the k_on_ and k_off_ estimates, we found a good correlation for all four agonists at both the WT and S193^5.42^A mutant receptors. However, the kinetic K_d_ values were found to systematically overestimate the potency/affinity of the tested agonists (see Table 1). Although the *Y*-axis intercepts of the k_obs_—[agonist] plots were generally in good agreement with the k_off_ estimates obtained from GIRK current deactivation time courses, plotting the Y intercepts against k_off_s from washout experiments reveals a tendency of the former values to be higher (Appendix A). Thus, we cannot exclude the possibility that buffer exchange limited our k_off_ estimates, leading to the overestimation of ligand potency. In agreement with this notion, we were previously able to record rates of DA-evoked GIRK current inhibition by competitive D_2_R antagonists of just above 0.4 s^−1^—twice as fast as the response decay rates observed here [28]. Finally, the rate of GTP hydrolysis at the G protein would, even in the presence of RGS4, restricted our ability to accurately quantify very rapid rates of agonist dissociation.

S193^5.42^ in D_2_R has consistently been reported to be crucial for high-affinity binding of DA and of (S)- but not (R)-5-OH-DPAT, nor of p-tyramine [9,14,15]. This notion is supported by the electrophysiology data and molecular dynamics simulations presented here. Accordingly, the simulated binding modes for DA and (S)-5-OH-DPAT revealed prominent polar contacts with S193^5.42^, and S193^5.42^A mutation was found to decrease experimentally determined k_on_ estimates and potencies for both DA and (S)-5-OH-DPAT. Conversely, the mutation marginally increased (R)-5-OH-DPAT k_on_, such that the k_on_s of both enantiomers were similar at the mutant receptor. S193^5.42^A mutation slightly increased p-tyramine k_on_ while having a negligible effect on k_off_. Somewhat surprisingly, DA k_off_ was also unaffected by S193^5.42^A mutation. This finding should, however, be interpreted with caution due to the limiting effects of buffer exchange and GTP hydrolysis noted above and the fact that the DA k_off_ estimates were the fastest in the dataset. Thus, the apparent lack of effect of S193^5.42^A mutation on DA k_off_ may be due to insufficient temporal resolution in our experimental system. Alternatively, there may indeed be no appreciable effect of the mutation on DA k_off_, putatively due to the receptor-agonist-G protein complex undergoing isomerization after initial agonist binding and receptor activation, such that S193^5.42^A no longer forms important contacts with the agonist.

Furthermore, the mutation also increased the relative efficacies of (R)-5-OH-DPAT and p-tyramine, while the relative efficacy of (S)-5-remained virtually unchanged. However, since agonist efficacy was determined by normalization to the maximal DA-induced response, another possible interpretation would be that the efficacies of DA and (S)-5-OH-DPAT decreased, while those of (R)-5-OH-DPAT and p-tyramine were less affected. The effects of S193^5.42^A mutation on all four agonists are summarized in Table 2.

Interestingly, the k_off_s of (R)- and (S)-5-OH-DPAT were similar at the WT D_2_R and were affected by mutation to a similar extent despite their differential interactions with S193^5.42^. It thus appears that interaction with S193^5.42^ may slow dissociation of both (R)- and (S)-5-OH-DPAT from the receptor. As expected, the results of our molecular dynamics simulations suggest that the main binding mode of (R)-5-OH-DPAT is not altered by the S193^5.42^A mutation (Figure 4C,D). Thus, the observed shifts of (R)-5-OH-DPAT kinetics appear to be mediated by a change in the ligand entry/exit pathway. Indeed, a detailed analysis of the simulation data of (R)-5-OH-DPAT in complex with the WT D_2_R suggests that apart from the main binding mode involving S197^5.46^ (Figure 5A,B, in green), the existence of an additional meta-stable binding mode (Figure 5A,C, in red). As this binding mode depends on polar contacts between (R)-5-OH-DPAT and S193^5.42^, it would be absent in the S193^5.42^A mutant receptor. We speculate that the meta-stable binding mode forms a step in the binding/unbinding process of (R)-5-OH-DPAT. Before dissociating from D_2_R, (R)-5-OH-DPAT would thus assume the meta-stable conformation, from which it would either proceed to complete dissociation, or return to the main binding mode (Figure 5D, left panel). Hence, the presence of this meta-stable binding mode could slow ligand dissociation. On the other hand, the lack of the meta-stable binding mode in the S193^5.42^A mutant D_2_R would permit fast exchange between the bound and unbound conformations, which would explain the increase in (R)-5-OH-DPAT k_off_ observed at the S193^5.42^A mutant (Figure 5D, right panel).

Interactions with S193^5.42^ have long been known to be crucial for affinities and functional potencies of several D_2_R agonist ligands. Unexpectedly, our results from GIRK channel activation experiments suggest that these contacts may have a greater impact on agonist association rate constants (k_on_s) rather than on agonist dissociation rate constants (k_off_s). Optimization of therapeutic or diagnostic ligand binding kinetics is important for optimal target engagement in vivo [19,20,49]. With the advantage of using unmodified proteins, ligands and living cells, the approach presented here may therefore be of interest for estimation of binding kinetics of drug candidates at the D_2_R and other GPCRs capable of activating GIRK channels.

## 4. Materials and Methods

### 4.1. Molecular Biology

cDNA encoding the wildtype (WT) and S193^5.42A^A human dopamine D_2S_ (short isoform; from Dr. Marc Caron, Duke University, NC, USA) receptors were in pXOOM (provided by Dr. Søren-Peter Olesen, University of Copenhagen, Denmark) whereas RGS4 (from the cDNA Resource Center, Bloomsburg, PA, USA; www.cdna.org, access date 1 March 2021) and GIRK1/4 (a gift from Dr. Terence Hebert, McGill University, Montreal, QC, Canada) were in pcDNA3.1+. The S193^5.42^A point mutation of the D_2_R was made using the QuickChange Lightning kit (Agilent Technologies, Santa Clara, CA, USA), according to the manufacturer’s instructions, and confirmed by DNA sequencing of the entire insert. Plasmids were linearized using suitable restriction enzymes (WT D_2_R, S193^5.42^A mutant D_2_R, and RGS4: XhoI; GIRK1 and GIRK4: NotI), followed by in vitro transcription using the T7 mMessage mMachine kit (Ambion, Austin, TX, USA). cRNA concentration and purity were determined by spectrophotometry.

### 4.2. Oocyte Preparation

Oocytes from the African clawed toad, *Xenopus laevis*, were surgically isolated as described previously [12]. The procedure was approved by the Swedish National Board for Laboratory Animals and the Animal Welfare Ethical Committee in Stockholm (approval number N245/15). Following one day of incubation at 12 °C, oocytes were injected with 50 nL containing 0.2 ng D_2_R cRNA, 40 ng of RGS4, and 1 ng of each GIRK1 and GIRK4 cRNA using the Nanoject II (Drummond Scientific, Broomall, PA, USA).

### 4.3. Electrophysiological Methods

Injected cells were incubated for another 6 days at 12 °C in modified Barth’s solution (MBS), composed of (in mM): 88 NaCl, 1 KCl, 2.4 NaHCO_3_, 15 HEPES, 0.33 Ca(NO_3_)_2_, 0.41 CaCl_2_, 0.92 MgSO_4_, and 2.5 sodium pyruvate, supplemented with 25 U/mL penicillin and 25 µg/mL streptomycin and adjusted to pH 7.6 with NaOH. Electrophysiological recordings were performed at 22 ℃ using the eight-channel, two-electrode voltage-clamp OpusXpress 6000A (Molecular Devices, San José, CA, USA) [50]. Continuous perfusion was maintained at either 1 (concentration–response experiments) or 4.5 (for estimation of kinetic parameters) mL/min. Data were acquired at membrane potentials of −80 mV and sampled at 156 Hz using the OpusXpress 1.10.42 (Molecular Devices) software. To increase the inward rectifier potassium channel current at negative potentials, a high-potassium extracellular buffer was used (in mM: 64 NaCl, 25 KCl, 0.8 MgCl_2_, 0.4 CaCl_2_, 15 HEPES, and 1 ascorbic acid, adjusted to pH 7.4 with NaOH), yielding a K^+^ reversal potential of about −40 mV. In experiments with 1 mM KCl buffer, the NaCl concentration was adjusted to 88 mM. Ascorbic acid was included to prevent the spontaneous oxidation of DA. For concentration-response experiments, each oocyte was first exposed to 1 µM (WT) or 300 µM (S193^5.42^A) DA, evoking a maximal D_2_R-mediated response. After washout of DA, an initial stabilization period of 60 s where 25 KCl buffer was perfused was followed by applications of three to five increasing concentrations of agonist, which were added consecutively at 60 s intervals (see Appendix A for a representative current trace showing the responses to increasing concentrations of p-tyramine). For each concentration of agonist, the relative current amplitude after 60 s (when a steady-state response had been achieved) of agonist perfusion was plotted to generate the concentration-response relationships. Oocytes were selected for electrophysiology recordings based on having holding currents at −40 mV of less than 0.5 µA. Likewise, recordings where holding currents at −40 mV were greater than 0.5 µA after data acquisition at −80 mV were discarded.

### 4.4. Ligands

DA and p-tyramine were from Sigma-Aldrich (St. Louis, MO, USA) and were prepared fresh on each day of experiments and dissolved directly in the recording buffer. (R)- and (S)-5-OH-DPAT (Axon MedChem BV, Groeningen, The Netherlands) were dissolved in DMSO at 10 mM and subsequently diluted into the recording buffer at the desired concentrations. Likewise, haloperidol (Abcam Chemicals, Cambridge, UK) was dissolved at 10 mM in DMSO and diluted in recording buffer to the final concentration. The maximum final concentration of DMSO used in any experiment was 0.1% *v/v*.

### 4.5. Data Analysis

Electrophysiology concentration-response data was initially processed in Clampfit 10.6 (Molecular Devices) by subtracting the basal (agonist-independent) current and quantifying the current amplitude evoked by each concentration of agonist. Agonist concentration–response relationships were analyzed by fitting sigmoidal functions using nonlinear regression in GraphPad Prism 8 (GraphPad Software, San Diego, CA, USA). The following equation was used for fitting:Y = Top/(1 + 10 ^(LogEC50 − X)^)(1)
where Y is the GIRK current response normalized to the response to a maximally effective concentration of DA, Top is the maximal response of the agonist in question, and X is the logarithm of agonist concentration. Top and its SEM were used to plot the efficacy values shown in Figure 1E.

Recordings of GIRK response deactivation were transferred to Matrix Laboratory 2018b (MathWorks, Natick, MA, USA), peak normalized and time-averaged. Data are reported as mean ± SEM throughout the manuscript. Monoexponential functions were fitted to the relevant intervals of individual traces corresponding to response decay following agonist washout (see Section 2), outputting the deactivation time constant, τ_deact_. For DA and p-tyramine, the first 42 s, and for (S)- and (R)-5-OH-DPATs, the first 104 s following agonist washout were used to fit the exponential functions at WT D_2_R. 10 nM DA, 10 nM (S)-5-OH-DPAT, 300 nM (R)-5-OH-DPAT, and 10 µM p-tyramine were used for these experiments at the WT receptor. At the S193^5.42^A mutant, 10 µM DA, 1 µM (S)-5-OH-DPAT, 300 nM (R)-5-OH-DPAT, and 1 mM p-tyramine were used, and the first 24 s of washout were used to fit the exponential functions for all agonists. Estimates of the dissociation rate constant, k_off_, were obtained as 1/τ_deact_.

Association rate constant estimates were derived from recordings of GIRK current activation in response to applications of various agonist concentrations. Monoexponential functions were fit to cover 80% of the current increase in response to agonist using Levenberg–Marquardt least-squares fitting in Clampfit 10, outputting the activation time constant, τ_act_. The observed activation rate (k_obs_) was defined as 1/τ_act_, and subsequently used for estimation of the association rate constant as the slope of the dependence of k_obs_ on agonist concentration, over the range of concentrations where this relation was linear (see also [28]), using the following relation:k_obs_ = [A] × k_on_ + k_off_(2)
where [A] is the agonist concentration, k_on_ the association rate constant, and k_off_ the dissociation rate. However, rather than using Equation (2), k_off_ was calculated separately from the response decay time constant upon agonist washout, τ_deact_, as described above.

Kinetic K_d_s were calculated as:K_d_ = k_off_/k_on_(3)

Statistical analysis was performed using GraphPad Prism 8. *p* < 0.05 was chosen as the significance limit.

### 4.6. Molecular Dynamics Simulations

The simulated complexes were generated by docking the studied compounds into the structure of D_2_R in an active conformation [PDB code: 6VMS] [3]. For docking, we used the standard docking method available in the MOE package (www.chemcomp.com, access date 1 March 2021). The poses were selected based on the docking score, as well as available mutational data [9,10,11,13,14]. The generated WT complex was oriented in the membrane using coordinates obtained from the OPM database (Positioning of proteins in membranes: a computational approach), and solvated with TIP3 waters, using the CHARMM-GUI server [51]. The ionic strength of the system was kept at 0.15 M using NaCl ions. The S193^5.42^A mutation was introduced using the CHARMM-GUI pipeline.

Simulations were carried out using the ACEMD simulation package [52]. Ligand parameters were assigned by ParamChem from the CGenFF force field [53,54]. Parameters for other system components were obtained from CHARMM36m [55] and CHARMM36 force fields [56]. In the simulation protocol, we adhere to the guidelines of the GPCRmd consortium [57].

The systems were first relaxed during 100 ns of simulations under constant pressure and temperature (NPT) with a time step of 2 fs, with harmonic constraints applied to the protein backbone and ligand heavy atoms. The temperature was maintained at 310 K using the Langevin thermostat [58] and pressure was kept at 1 bar using the Berendsen barostat [59]. The equilibration run was followed by two parallel 400 ns production runs in conditions of constant volume and temperature (NVT) with a 4-fs time step. No constraints were applied during this stage. In all simulations, we used van der Waals and short-range electrostatic interactions with a cutoff of 9 Å and the particle mesh Ewald method [60] for long-range electrostatic interactions. The resulting simulation frames were analyzed using VMD [61] and tools available within.

## Figures and Tables

**Figure 1 ijms-22-04078-f001:**
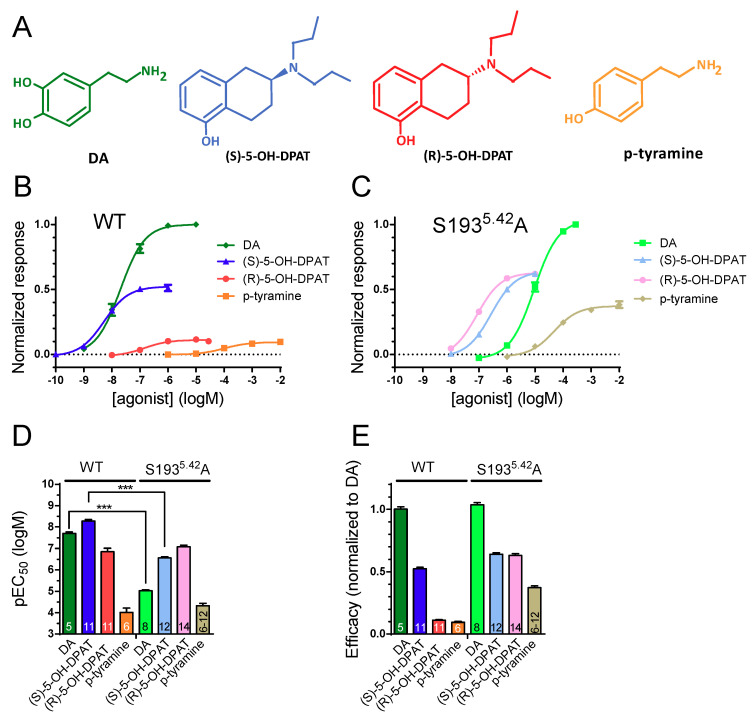
Concentration-response relationships of agonists at WT D_2_R and S193^5.42^A. Concentration–response curves for GIRK activation elicited by application of DA, (S)-5-OH-DPAT, (R)-5-OH-DPAT, and p-tyramine (structures shown in panel (**A**)) in oocytes co-expressing GIRK1/4 subunits and RGS4 with (**B**) WT D_2_R and (**C**) D_2_R S193^5.42^A. (**D**) Agonist pEC*_50_*s at WT and S193^5.42^A D_2_R: pEC_50_ for DA (WT: pEC_50_ = 7.70 ± 0.07, *n* = 5; S193^5.42^A: pEC_50_ = 5.03 ± 0.04, *n* = 8), (S)-5-OH-DPAT (WT: pEC_50_ = 8.28 ± 0.08, *n* = 11; S193^5.42^A: pEC_50_ = 6.56 ± 0.05, *n* = 12), (R)-5-OH-DPAT (WT: pEC_50_ = 6.85 ± 0.16, *n* = 11; S193^5.42^A: pEC_50_ = 7.07 ± 0.08, *n* = 14), and p-tyramine (WT: pEC_50_ = 4.00 ± 0.21, *n* = 6; S193^5.42^A: pEC_50_ = 4.32 ± 0.12, *n* = 6–12). Comparison of pEC*_50_*s using two-way ANOVA yielded significant main effects of agonist (F_(3, 65*)*_ = 281.8) and of the S193*^5.42^*A mutation (F_(1, 65)_ = 139.2), as well as a significant interaction between these two factors (F_(3, 65)_ = 79.98, *p* < 0.001 for each main effect). Sidak’s multiple comparisons test further revealed that the pEC*_50_*s of DA and (S)-5-OH-DPAT, but not p-tyramine and (R)-5-OH-DPAT, differed significantly between WT and mutant D*_2_*R, as indicated by asterisks; ***, *p* < 0.001. (**E**) Relative efficacies at WT D_2_R and S193^5.42^A for DA (WT: 1.00 ± 0.02; S193^5.42^A: 1.04 ± 0.04), (S)-5-OH-DPAT (WT: 0.52 ± 0.01; S193^5.42^A: 0.64 ± 0.01), (R)-5-OH-DPAT (WT: 0.11 ± 0.01; S193^5.42^A: 0.63 ± 0.02), and p-tyramine (WT: 0.09 ± 0.01; S193^5.42^A: 0.37 ± 0.01). WT and S193^5.42^A responses were normalized to the response evoked by 1 µM and 300 µM DA, respectively. The efficacy values were obtained from the fitted parameter Top, from the corresponding concentration-response curves (see Materials and Methods). The number of oocytes used for each condition is indicated on the bars in (**D**,**E**) and corresponds to the number recorded to generate the data points plotted in (**B**,**C**). Experiments were performed using a buffer perfusion rate of 1 mL/min. Data are presented as mean ± SEM.

**Figure 2 ijms-22-04078-f002:**
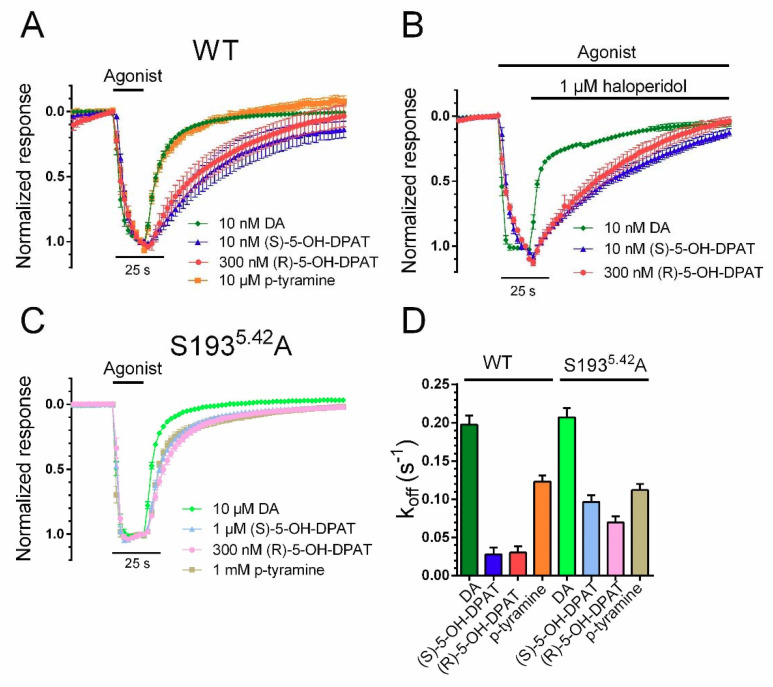
Kinetics of GIRK current deactivation following agonist washout was used to estimate agonist k_off_. (**A**) Response decay time courses following washout of DA, (S)-5-OH-DPAT, (R)-5-OH-DPAT, and p-tyramine from oocytes co-expressing WT D_2_R with GIRK1/4 and RGS4. (**B**) Terminating the agonist-induced response by application of 1 µM haloperidol revealed similar rates of decay as observed in the agonist washout experiments presented in (**A**). (**C**) Response decay time constants following washout of DA, (S)-5-OH-DPAT, (R)-5-OH-DPAT, and p-tyramine from oocytes co-expressing S193^5.42^A D_2_R with GIRK1/4 and RGS4. (**D**) Estimated dissociation rate constants at WT D_2_R and S193^5.42^A for DA (WT: 0.197 ± 0.012 s*^−^*^1^, *n* = 6; S193^5.42^A: 0.207 ± 0.012 s*^−^*^1^, *n* = 8), (S)-5-OH-DPAT (WT: 0.028 ± 0.010 s*^−^*^1^, *n* = 4; S193^5.42^A: 0.096 ± 0.009 s*^−^*^1^, *n* = 6), (R)-5-OH-DPAT (WT: 0.030 ± 0.008 s*^−^*^1^, *n* = 5; S193^5.42^A: 0.069 ± 0.008 s*^−^*^1^, *n* = 6), and p-tyramine (WT: 0.123 ± 0.008 s*^−^*^1^, *n* = 7; S193^5.42^A: 0.112 ± 0.008 s*^−^*^1^, *n* = 5), determined by fitting exponential functions to the agonist washout phases. For DA and p-tyramine, the first 42 s, and for (S)- and (R)-5-OH-DPATs, the first 104 s following agonist washout were used to fit the exponential functions at WT D_2_R. 10 nM DA, 10 nM (S)-5-OH-DPAT, 300 nM (R)-5-OH-DPAT, and 10 µM p-tyramine were used for these experiments at the WT receptor. At D_2_R S193^5.42^A, 10 µM DA, 1 µM (S)-5-OH-DPAT, 300 nM (R)-5-OH-DPAT, and 1 mM p-tyramine were used, and the first 24 s of the washout were used to fit the exponential functions for all agonists. Experiments were performed using a perfusion rate of 4.5 mL/min. Data are presented as mean ± SEM.

**Figure 3 ijms-22-04078-f003:**
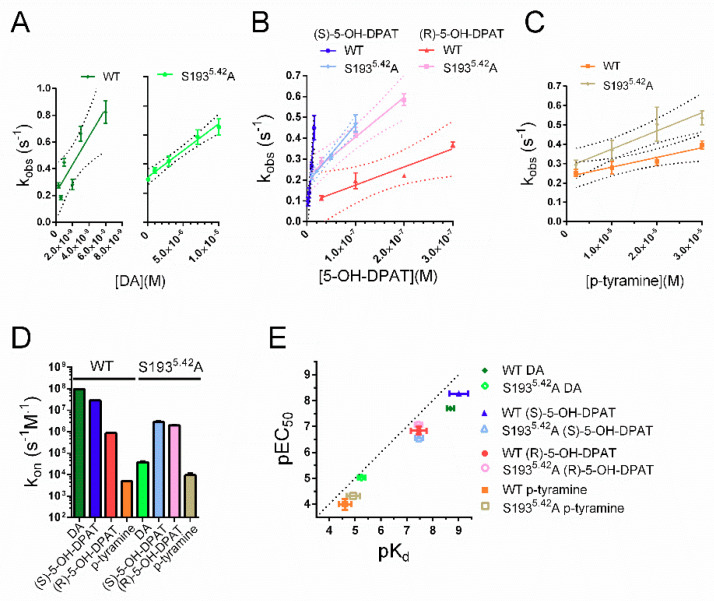
Estimation of agonist k_on_ at WT and S193^5.42^A mutant D_2_R. Individual concentrations of (**A**) DA, (**B**) (S)-5-OH-DPAT and (R)-5-OH-DPAT, and (**C**) p-tyramine were added to oocytes co-expressing WT D_2_R or D_2_R S193^5.42^A with RGS4 and GIRK1/4. The rates of rise (k_obs_) of the resulting current responses have been plotted against the corresponding agonist concentrations and linear fits (solid lines) and their 95% confidence bands (dotted lines) are shown. Experiments were performed using a perfusion rate of 4.5 mL/min. (**D**) Summary statistics for k_on_, estimated from the slopes of the linear fits to k_obs_ shown in (**A**), (**B**), and (**C**). Shown are mean k_on_ estimates for DA (WT: 9.70 ± 1.23 × 10^7^ s*^−^*^1^ × M*^−^*^1^, S193^5.42^A: 3.69 ± 0.48 × 10^4^ s*^−^*^1^ × M*^−^*^1^), (S)-5-OH-DPAT (WT: 2.86 ± 0.31 × 10^7^ s*^−^*^1^ × M*^−^*^1^, S193^5.42^A: 2.82 ± 0.34 × 10^6^ s*^−^*^1^ × M*^−^*^1^), (R)-5-OH-DPAT (WT: 8.65 ± 1.21 × 10^5^ s*^−^*^1^ × M*^−^*^1^, S193^5.42^A: 1.95 ± 0.15 × 10^6^ s*^−^*^1^ × M*^−^*^1^), and p-tyramine (WT: 4.94 ± 1.15 × 10^3^ s*^−^*^1^ × M*^−^*^1^, S193^5.42^A: 9.41 ± 2.34 × 10^3^ s*^−^*^1^ × M*^−^*^1^). Note the logarithmic y-axis. (**E**) Relations between kinetic pK_d_s, derived from k_off_ and k_on_, and pEC_50_s, obtained from the concentration–response experiments shown in Figure 1, at WT D_2_R and S193^5.42^A. The correlation between pEC_50_s and kinetic pK_d_s for all four agonists at both WT and S193^5.42^A mutant D_2_R was statistically significant (Spearman’s r = 0.9048, *p* = 0.0046) for all four pairs. Data are presented as mean ± SEM.

**Figure 4 ijms-22-04078-f004:**
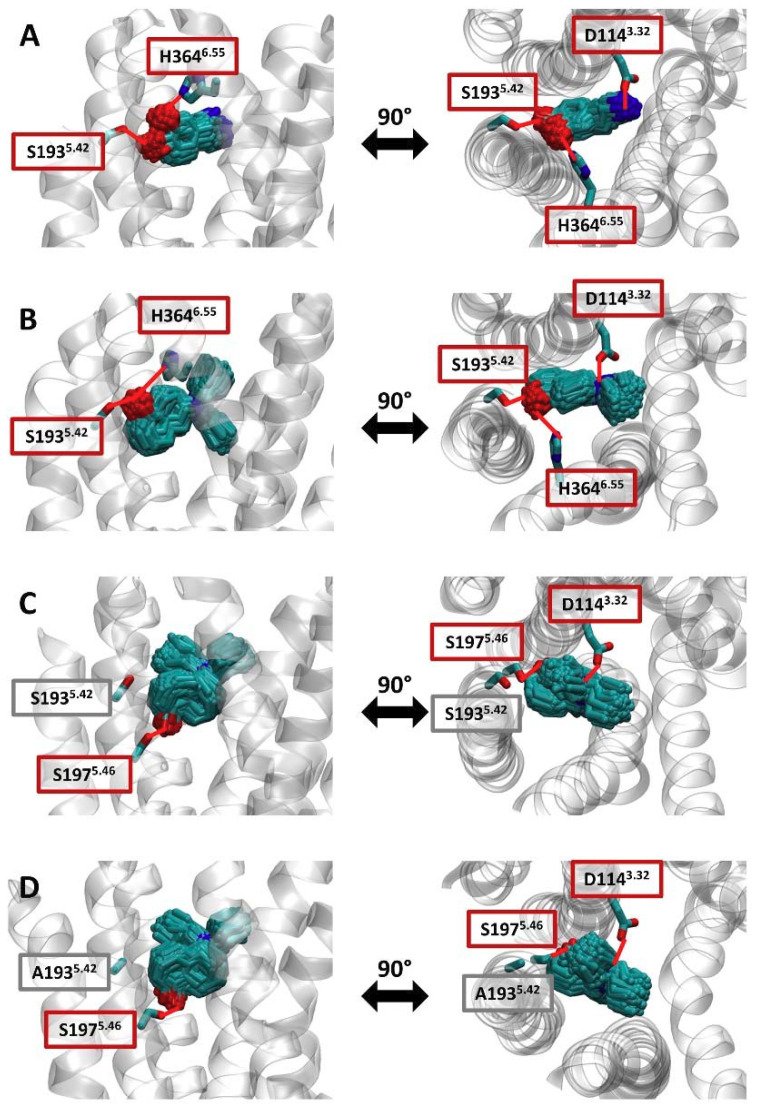
Binding modes of studied compounds within the WT D_2_R and the S193^5.42^A mutant receptor. Simulations of (**A**) DA, (**B**) (S)-5-OH-DPAT, and (**C**) (R)-5-OH-DPAT in complex with the WT D_2_R and (**D**) (R)-5-OH-DPAT in complex with the S193^5.42^A D_2_R were clustered based on the root-mean-square deviation (RMSD) of the ligand. For each of the systems, the poses of the ligand within the most populated cluster are depicted in licorice. For each of the binding modes, the studied position 193^5.42^, as well as the residues that formed polar interactions with the ligand, are also shown in licorice. Polar interactions are highlighted by red lines.

**Figure 5 ijms-22-04078-f005:**
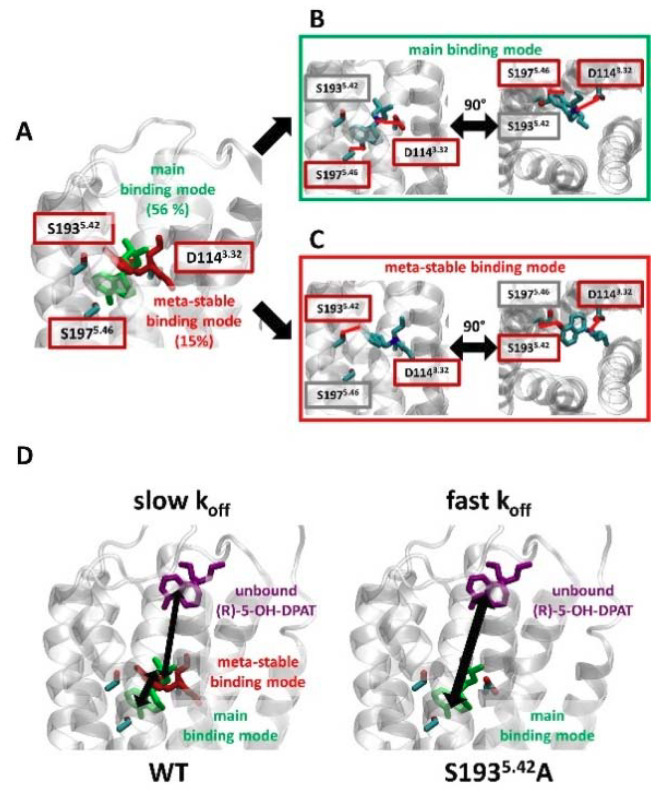
(R)-5-OH-DPAT forms a meta-stable binding mode with S193^5.42^. (**A**) Clustering simulations of (R)-5-OH-DPAT in complex with WT D_2_R, based on the RMSD of the ligand, reveal two binding modes. The main binding mode was maintained over 56% of the simulation frames (green) and a meta-stable binding mode was maintained over 15% of the simulation frames (red). (**B**,**C**). For each of the binding modes, the studied position S193^5.42^, as well as residues that form polar interactions with the ligand, are shown. These polar interactions are highlighted with red lines. (**D**) A model explaining the slow K_off_ values observed for the unbinding of (R)-5-OH-DPAT from WT D_2_R. Before dissociating from the receptor (purple conformation), (R)-5-OH-DPAT bound in the main binding mode (green) assumes a meta-stable binding mode (red). When bound in this meta-stable binding mode, (R)-5-OH-DPAT can revert to the main binding mode or proceed to an unbound conformation. Hence, the dissociation of (R)-5-OH-DPAT is effectively slowed down by S193^5.42^. In comparison, at the S193^5.42^A mutant receptor, the absence of the meta-stable binding mode permits fast exchange between the bound and unbound conformations of (R)-5-OH-DPAT.

**Table 1 ijms-22-04078-t001:** Agonist EC_50_s, estimated forward (k_on_) and reverse (k_off_) rate constants, and kinetic K_d_s at WT and S193^5.42^A mutant D_2_R.

	pEC_50_ ± SEM (EC_50_, nM)	N	k_off_ ± SEM (s^−1^)	N	k_on_ ± SEM (s^−1^ × M^−1^)	N	pK_d_ ± SEM
**WT**							
DA	7.70 ± 0.07 (20)	5	0.197 ± 0.012	6	9.70 ± 1.23 × 10^7^	3–8	8.69 ± 0.14
p-tyramine	4.00 ± 0.21 (100,000)	6	0.123 ± 0.008	7	4.94 ± 1.15 × 10^3^	5–7	4.61 ± 0.24
(S)-5-OH-DPAT	8.28 ± 0.08 (5)	11	0.028 ± 0.010	4	2.86 ± 0.31 × 10^7^	5–9	9.01 ± 0.35
(R)-5-OH-DPAT	6.85 ± 0.16 (141)	11	0.030 ± 0.008	5	8.65 ± 1.21 × 10^5^	3–8	7.46 ± 0.30
**S193^5.42^A**							
DA	5.03 ± 0.04 (9333)	8	0.207 ± 0.012	8	3.69 ± 0.48 × 10^4^	4–7	5.25 ± 0.14
p-tyramine	4.32 ± 0.12 (47,863)	6–12	0.112 ± 0.008	5	9.41 ± 2.34 × 10^3^	4–6	4.93 ± 0.26
(S)-5-OH-DPAT	6.56 ± 0.05 (275)	12	0.096 ± 0.009	6	2.82 ± 0.34 × 10^6^	5–8	7.47 ± 0.15
(R)-5-OH-DPAT	7.07 ± 0.08 (85)	14	0.069 ± 0.008	6	1.95 ± 0.15 × 10^6^	4–7	7.45 ± 0.14

**Table 2 ijms-22-04078-t002:** Relative changes in agonist potencies, forward (k_on_) and reverse (k_off_) rate constants and calculated kinetic potencies at the S193^5.42^A mutant, as compared to WT D_2_R.

Agonist	Potency	Relative Efficacy	k_off_	k_on_
DA	↓↓↓	−	−	↓↓↓
p-tyramine	↑	↑	−	↑
(S)-5-OH-DPAT	↓↓	−	↑	↓↓
(R)-5-OH-DPAT	↑	↑↑	↑	↑

↑↑, strong increase; ↑, weak increase; ↓↓↓, very strong decrease; ↓↓, strong decrease.

## Data Availability

The data presented in this study are available in the article and in the Appendix A.

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
