# Peer review of "Dopamine D_2_ Receptor Agonist Binding Kinetics—Role of a Conserved Serine Residue"

_ijms, 2021, doi:10.3390/ijms22084078_

Round 1
Reviewer 1 Report
Dear Authors, according to my experience, the manuscript you presented for publication is generally well written and the study well organized. Please find some issues should be addressed. Introduction: Lines 52-53: Please check the first mention of the term “transmembrane” and move the first abbreviation accordingly. Same for amino acids letters code throughout the text Lines 55-56: Please also introduce physiological relevance and location of D2 receptors beyond pathological conditions. For example, D2 can be either postsynaptic (expressed by the striatum medium spiny neurons) or presynaptic (expressed in dopamine neurons terminals). Mentioning D2 subtypes (D2-D3-D4) and heterodimers with other receptors has relevance in your study as well. Stating that “D2 receptor (D2R) couples to inhibitory Gi/o proteins” is not enought. Please mention diversity of downstream signalling, which could be relevant for your purpose. Finally, please specify any possible involvement of the Serine/mutation in the above mentioned points. Results/Discussion: Have the 4 agonists been applied to the same oocyte following washout? If so, which is the order of application for the agonist and the rationale for the choice. In the alternative case “one agonist each oocyte”: currents could be influenced by different expression levels of D2 and/or GIRK in different cells. In this regards, please also spcify the number of oocytes for each experiments (for instance in line 139) Several papers show direct activity of haloperidol at GIRK channels (for example Kobayashi et al 2000 DOI: 10.1038/sj.bjp.0703224). Authors should clearly state (better if any data can be presented) that the concentration used in this study is ineffective onto GIRK.Author Response
Reviewer 1:
Dear Authors, according to my experience, the manuscript you presented for publication is generally well written and the study well organized. Please find some issues should be addressed.
Introduction: Lines 52-53: Please check the first mention of the term “transmembrane” and move the first abbreviation accordingly. Same for amino acids letters code throughout the text
Our response: This has now been corrected (lines 52, 54, 73, 286).
Lines 55-56: Please also introduce physiological relevance and location of D2 receptors beyond pathological conditions. For example, D2 can be either postsynaptic (expressed by the striatum medium spiny neurons) or presynaptic (expressed in dopamine neurons terminals). Mentioning D2 subtypes (D2-D3-D4) and heterodimers with other receptors has relevance in your study as well. Stating that “D2 receptor (D2R) couples to inhibitory Gi/o proteins” is not enought. Please mention diversity of downstream signalling, which could be relevant for your purpose. Finally, please specify any possible involvement of the Serine/mutation in the above mentioned points.
Our response: We have now included information about the physiological functions and locations of D2 receptors, dopamine receptor subtypes, receptor heteromerization, diversity of downstream signaling pathways (lines 55-70) and the differential impact of S1935.42A mutation on the induction of some signaling pathways by certain synthetic agonists (lines 81-83).
Results/Discussion: Have the 4 agonists been applied to the same oocyte following washout? If so, which is the order of application for the agonist and the rationale for the choice. In the alternative case “one agonist each oocyte”: currents could be influenced by different expression levels of D2 and/or GIRK in different cells. In this regards, please also spcify the number of oocytes for each experiments (for instance in line 139)
Our response: Concentration-response data was obtained for only one agonist per oocyte, in order to avoid oocyte deterioration and GIRK current rundown during prolonged registration. This is now specified in the manuscript (lines 132-137). The number of oocytes used are the same for panels B-E and are defined in the legend of Figure 1. The numbers are now also shown on the bar graphs, for clarity. We do not believe that the relative efficacies were influenced much by different expression levels, since there is little or no receptor reserve under the conditions used (amount of D2 receptor RNA and RGS4 coexpression), as is now mentioned in the text (lines 147-153). In agreement, the relative sizes of mean GIRK current amplitudes evoked by maximally effective concentrations of each agonist across all experiments (see new Supplementary Figure S2) are similar to the relative efficacies shown in Figure 1E.
Several papers show direct activity of haloperidol at GIRK channels (for example Kobayashi et al 2000 DOI: 10.1038/sj.bjp.0703224). Authors should clearly state (better if any data can be presented) that the concentration used in this study is ineffective onto GIRK.
Our response: Extrapolation of the concentration-response curve presented by Kobayashi et al. (2000) suggests that haloperidol inhibition of GIRK1/4 currents is very low (~2%) at 1 µM. The lowest concentration tested by Kobayashi et al. is 3 µM. In agreement, Heusler et al. (2007) reported an absence of GIRK blockade at 1 µM haloperidol. We have also measured direct GIRK block by 3 µM haloperidol under the conditions used here and found this block to be ~5.9%. The antagonist effect at the D2 receptor is much more potent; we have previously found 1 µM haloperidol to fully antagonize the response to 100 nM dopamine (Sahlholm et al., 2016, Eur Neuropsychopharmacol. DOI: 10.1016/j.euroneuro.2016.01.001 ). Thus, the major effect of haloperidol observed here is most likely due to D2 receptor inhibition resulting from competition with dopamine. This information is now included in the manuscript (lines 201-208).
Reviewer 2 Report
Overall, the work presented here is a necessary step toward a better understanding of the contribution of the conserved serine residue S1935.42A in the association/dissociation dynamics of D2R. This work also presents an interesting approach that provides a connection between pharmacological occupancy studies (i.e. radioligand binding), and the actual physiology by using an electrophysiological approach. The methods are sound, and the authors do a nice job presenting the work within the context of its relevance, but I have some comments that would be valuable to address to further clarify the data and the findings presented.
Results
2.1 Phenethylamine and DPAT potencies and efficacies at WT and S1935.42A D2R
Was the same oocyte treated with increasing concentrations of the drug in question? I am assuming was not, but there is no mention of this in the methods section, if so the timeline and detailed method for this should be described. Assuming was not, for how long was allowed to perfuse every concentration used after stablishing a stable recording? Also, how many cells were recorded for every data point used for the fitting?
Fig. 1
Panel B and C. As stated above, there is not statement of how many cells were recorded to generate the normalized curves (data points). This should be stated in the caption at least, seems to be such tight error bars (SEM) in every point used to generate the fitted curve. Would be valuable to see absolute values of GIRK currents to have a better idea of how the response can vary between cells. Would be probably also valuable to provide a representative trace of the currents measured along with the protocol employed.
Figure 1 Caption. It is stated that (d) is the summary of statistics of WT D2R and S1935.42A pEC50 for the different drugs tested. What type of statistics were used? Seems to be some differences but without knowing the sample size is hard to know. Was this refereeing only to the column descriptive statistics? If so, wouldn’t be valuable to add some factorial analysis to identify sources of variation?
Methods:
4.3. Electrophysiological methods
It appears to be a lot of missing information in this section, although a previous work was cited, at least a brief description of procedure and timelines should be stated here for the better understanding of this work.
Please describe the timeline for electrophysiological recordings. Were any steps taken to determine the stability of the recordings? Are the authors confident that the recordings were stable at baseline and after drug superfusion? Were any analyses done to confirm that parameters for each recording were stable in the presence or absence of drugs? For the kinetic analyses, a rate of 4.5mL/min is quite high, I wonder what steps were taken to be make sure the recordings were stable or if there might be fluctuations in the measured currents that could be contributing to the changes observed.
How long was the drug was applied? Did the period of analysis begin at a specific time after onset of drug application, or did the timing of the analysis period vary with respect to the start of drug application? In the case of the former, at what time relative to onset of drug application was this? In the case of the latter, how was the analysis period selected? This would help to better understand the data provided and the limitations of this approach for the kinetic analysis.
Section 2.4 Structural determinants of mutation-induced changes in compound kinetics:No critiques or concerns here, just wanted to compliment the authors on a really nice job in describing and outlining the impact of agonist-receptor interactions associated with the mutation and contextualizing their findings in regard to the broader pharmacological and functional implications.
Minor.
Discussion.
Would be beneficial to discuss in more detail why the mutation did not impact DA koff. Addressing which other possibilities in the context of heterologous systems could be contributing to this lack of differences.
Author Response
Reviewer 2:
Overall, the work presented here is a necessary step toward a better understanding of the contribution of the conserved serine residue S1935.42A in the association/dissociation dynamics of D2R. This work also presents an interesting approach that provides a connection between pharmacological occupancy studies (i.e. radioligand binding), and the actual physiology by using an electrophysiological approach. The methods are sound, and the authors do a nice job presenting the work within the context of its relevance, but I have some comments that would be valuable to address to further clarify the data and the findings presented.
Results
2.1 Phenethylamine and DPAT potencies and efficacies at WT and S1935.42A D2R
Was the same oocyte treated with increasing concentrations of the drug in question? I am assuming was not, but there is no mention of this in the methods section, if so the timeline and detailed method for this should be described. Assuming was not, for how long was allowed to perfuse every concentration used after stablishing a stable recording? Also, how many cells were recorded for every data point used for the fitting?
Our response: The same oocytes were treated with increasing agonist concentrations, applied sequentially. Concentration-response data was acquired for only one agonist for each oocyte. Each concentration was applied for 60 s. For each concentration of agonist, the current amplitude after 60 s of agonist perfusion has been plotted to generate the concentration-response relationships. The protocol is now detailed in the Materials and Methods section (lines 468-479). The number of oocytes used is the same for panels B-E. This information is now stated more clearly in the legend of Figure 1, as well as on the bars in panels D and E of the same figure.
Fig. 1
Panel B and C. As stated above, there is not statement of how many cells were recorded to generate the normalized curves (data points). This should be stated in the caption at least, seems to be such tight error bars (SEM) in every point used to generate the fitted curve. Would be valuable to see absolute values of GIRK currents to have a better idea of how the response can vary between cells. Would be probably also valuable to provide a representative trace of the currents measured along with the protocol employed.
Our response: The number of oocytes used are now defined more clearly in the legend of Figure 1, as well as on the bars in panel D and E of the same figure. We have now added information about the absolute values of GIRK currents and a representative trace recorded with the protocol used to generate concentration-response curves (Supplementary Figure S1 and S2).
Figure 1 Caption. It is stated that (d) is the summary of statistics of WT D2R and S1935.42A pEC50 for the different drugs tested. What type of statistics were used? Seems to be some differences but without knowing the sample size is hard to know. Was this refereeing only to the column descriptive statistics? If so, wouldn’t be valuable to add some factorial analysis to identify sources of variation?
Our response: We were referring simply to descriptive statistics, but we have now performed a two-way ANOVA with genotype (i.e., WT or S1935.42A mutant receptor) and agonist as factors. Besides significant main effects of both mutation and agonist, we found a significant interaction between these two factors. Multiple comparisons revealed that the pEC50s for dopamine and (S)-5-OH-DPAT are significantly different between WT and S1935.42A mutant D2R. The results of the two-way ANOVA are stated in the legend of Figure 1 and asterisks have been added to panel D in the same figure, as appropriate.
Methods:
4.3. Electrophysiological methods
It appears to be a lot of missing information in this section, although a previous work was cited, at least a brief description of procedure and timelines should be stated here for the better understanding of this work.
Our response: The electrophysiological methods are now described in greater detail (lines 460-461 and 468-479; see our answer below).
Please describe the timeline for electrophysiological recordings. Were any steps taken to determine the stability of the recordings? Are the authors confident that the recordings were stable at baseline and after drug superfusion? Were any analyses done to confirm that parameters for each recording were stable in the presence or absence of drugs? For the kinetic analyses, a rate of 4.5mL/min is quite high, I wonder what steps were taken to be make sure the recordings were stable or if there might be fluctuations in the measured currents that could be contributing to the changes observed.
Our response: Initially, 60 s of buffer perfusion at -80 mV was performed to stabilize the baseline. This was followed by five consecutive 60 s agonist applications of increasing concentrations. For all assays, only oocytes showing holding currents of less than 0.5 µA at -40 mV were selected for experiments. Recordings from cells showing holding currents greater than 0.5 µA at -40 mV before or after recording at -80 mV were discarded. This information has now been added to the Materials and Methods section, as specified above.
How long was the drug was applied? Did the period of analysis begin at a specific time after onset of drug application, or did the timing of the analysis period vary with respect to the start of drug application? In the case of the former, at what time relative to onset of drug application was this? In the case of the latter, how was the analysis period selected? This would help to better understand the data provided and the limitations of this approach for the kinetic analysis.
Our response: The drug concentrations were applied for 60 s to reach a steady-state response, which was used for dose-response determinations. The intervals used for fitting of exponential functions for koff estimates were already mentioned in the legend of Figure 2 but have now been added to the Materials and Methods section as well (lines 506-511). For kobs estimation, exponential functions were fit to cover 80 % of the current increase in response to agonist. This information is now also included in the Materials and Methods section (line 515).
Section 2.4 Structural determinants of mutation-induced changes in compound kinetics:No critiques or concerns here, just wanted to compliment the authors on a really nice job in describing and outlining the impact of agonist-receptor interactions associated with the mutation and contextualizing their findings in regard to the broader pharmacological and functional implications.
Our response: We thank the reviewer for this positive comment.
Minor.
Discussion.
Would be beneficial to discuss in more detail why the mutation did not impact DA koff. Addressing which other possibilities in the context of heterologous systems could be contributing to this lack of differences.
Our response: We assume that either, there is no real difference, or the temporal resolution of our assay system is too limited to detect such a difference. Two sentences to this effect have been added to the discussion (lines 367-372).
Round 2
Reviewer 2 Report
The authors have addressed all my comments, and I do not have further concerns.